# Biomarkers of Spinal Cord Injury in Patients Undergoing Complex Endovascular Aortic Repair Procedures—A Narrative Review of Current Literature

**DOI:** 10.3390/biomedicines11051317

**Published:** 2023-04-28

**Authors:** Anna Sotir, Johannes Klopf, Christine Brostjan, Christoph Neumayer, Wolf Eilenberg

**Affiliations:** Division of Vascular Surgery, Department of General Surgery, Medical University of Vienna, 1090 Vienna, Austria; anna.sotir@meduniwien.ac.at (A.S.); johannes.klopf@meduniwien.ac.at (J.K.); christine.brostjan@meduniwien.ac.at (C.B.); christoph.neumayer@meduniwien.ac.at (C.N.)

**Keywords:** endovascular aortic repair, biomarkers, spinal cord ischemia, ischemia reperfusion injury, complex abdominal aortic aneurysm, thoracoabdominal aortic aneurysm, neuron specific enolase (NSE), glial fibrillary acidic protein (GFAP), neurofilament light (NFL), S100 calcium-binding protein β

## Abstract

Complex endovascular aortic repair (coEVAR) of thoracoabdominal aortic aneurysms (TAAA) has greatly evolved in the past decades. Despite substantial improvements of postoperative care, spinal cord injury (SCI) remains the most devastating complication of coEVAR being associated with impaired patient outcome and having an impact on long-term survival. The rising number of challenges of coEVAR, essentially associated with an extensive coverage of critical blood vessels supplying the spinal cord, resulted in the implementation of dedicated SCI prevention protocols. In addition to maintenance of adequate spinal cord perfusion pressure (SCPP), early detection of SCI plays an integral role in intra- and postoperative patient care. However, this is challenging due to difficulties with clinical neurological examinations during patient sedation in the postoperative setting. There is a rising amount of evidence, suggesting that subclinical forms of SCI might be accompanied by an elevation of biochemical markers, specific to neuronal tissue damage. Addressing this hypothesis, several studies have attempted to assess the potential of selected biomarkers with regard to early SCI diagnosis. In this review, we discuss biomarkers measured in patients undergoing coEVAR. Once validated in future prospective clinical studies, biomarkers of neuronal tissue damage may potentially add to the armamentarium of modalities for early SCI diagnosis and risk stratification.

## 1. Introduction

Endovascular aortic repair with complex stent-grafts (coEVAR) has become a valid alternative to open surgical repair (OSR), demonstrating lower perioperative mortality and morbidity rates with good mid- and long-term outcomes [1,2,3,4]. Owing to refinements in technical approaches and a steep learning curve for coEVAR, the number of cases is increasing and the indications are becoming wider [5].

Despite substantial improvements in perioperative care, spinal cord injury (SCI) until present remains the most devastating complication for the patient, as it is associated with dismal prognosis and short long-term survival [6,7]. A thorough understanding of the procedure and the anatomy of spinal cord blood supply reveals several risk factors for SCI in the context of coEVAR [8]. Amongst them, it is the extent of aortic coverage (>50 mm above the celiac trunk) and the presence of an occluded collateral network, that are associated with the highest risk of either transient or permanent neurological impairment [6,7,9,10,11]. Apart from general pathological mechanisms of ischemia reperfusion injury, an alternative theory suggests that embolization of an atheroma or thrombus arising from intra-aortic manipulations in addition to hypoperfusion of the spinal cord (SC) secondary to occlusion of segmental arteries (SAs) are involved in stent-graft associated progressive SCI formation [12]. 

Thus, rising challenges of coEVAR, implying a more extensive aortic coverage, has resulted in the implementation of vital clinical protocols designed to reduce the risk of postoperative SCI [5]. Essentially, clinical management of SCI consists of measures directed towards adequate maintenance of spinal cord perfusion pressure (SCPP) [8,13]. This is achieved either through an increase in mean arterial pressure (MAP), or through a decrease in lumbar pressure through cerebrospinal fluid drainage (CSFD). Based on concerted efforts of research and clinical teams, the indication for preoperative CSFD placement has been substantiated for high-risk patients for both thoracic (TEVAR) and coEVAR [14].

Early detection of SCI allows for active adjustment of SCPP when needed, before ischemic injury leads to irreversible damage of the nerve cells and tracts, resulting in persisting paraplegia or paraparesis [15]. The monitoring of SCI is challenging however, due to difficulties in clinical neurological examinations during patient sedation in the postoperative setting [5]. Considering this, increased attention has been given recently to establish a diagnostic modality, which would not only detect subclinical forms of SCI but would also represent a sensitive parameter for SCI risk stratification [16]. 

Ischemic damage of nerve tracts in the spinal cord has been shown to be accompanied by an elevation of proteins and metabolites from nervous tissue [17,18,19]. Therefore, several studies have investigated biomarkers as one of the most promising strategies for early SCI diagnosis [16,20,21]. 

In this narrative review, we discuss the published literature on the diagnostic potential of biomarkers of SCI in the context of coEVAR. The cohort of interest included patients treated for complex abdominal aortic aneurysms (coAAA) or TAAA with thoracic, fenestrated and/or branched stent-grafts (T/F/B-EVAR). The PubMed database was used to perform the literature search. The search string included the combination of the following keywords: “spinal cord injury”, “biomarkers”, “biochemical markers”, “thoracoabdominal aortic aneurysm”, “complex abdominal aortic aneurysm”, “endovascular aortic repair”, “fenestrated endovascular aortic repair”, “branched endovascular aortic repair”, “S100 β”, “GFAP”, “NFL”, “Tau” and “lactate”.

## 2. Markers of Stent-Graft Associated SCI

### 2.1. Markers of Astrocytic Activation

#### 2.1.1. S100 β (S100 Calcium-Binding Protein β)

S100 β has sparked a great deal of interest as a biochemical marker unique for central nervous system damage, released primarily during the acute phase of either traumatic or ischemic SCI [19]. As a member of the calcium-binding family proteins, S100 β is derived from the neural crest, and is predominantly expressed in glial and Schwann cells [16]. In the developing CNS, it acts as a neurotrophic factor and neuronal survival protein, whereas in the adult CNS the function of S100 β may be related to neurite extension, astrocytosis and axonal sprouting processes [22,23].

With regard to coEVAR procedures, the biomarker S100 β was measured in three studies. A pilot study by Nandate et al. assessed cerebrospinal fluid (CSF) levels of S100 β in ten patients undergoing TEVAR, none of which experienced a postoperative neurological deficit or changes in neurological examinations compared to the preoperative status [24]. S100 β levels immediately after stent-graft deployment and at 48 h did not reveal a significant difference compared to baseline levels. However, the concentration of S100 β was significantly increased in all patients at 24 h after the deployment of the stent-graft in comparison to the preoperative levels (*p* = 0.007). Following the dynamics of S100 β, the authors suggest that most of the patients undergoing TEVAR procedures might experience a subclinical form of mild SCI, which in general resolves and usually does not lead to clinically relevant neurological pathologies. 

A prospective clinical study by Brunnekreef et al. measured the level of S100 β in CSF at eight different time-points in elective TEVAR procedures in eight patients classified as high-risk for postoperative SCI [25]. While no central or peripheral neurological deficits were detected in the immediate postoperative phase, S100 β levels were not significantly elevated in comparison to baseline values. Yet, in one of the patients who was considered to have the highest risk for SCI, S100 β concentrations at 20 h after stent-graft placement were increased by more than three-fold in comparison to baseline (2.5 µg/L vs. <0.7 µg/L). Considering the evidence of studies assessing S100 β levels in patients undergoing OSR, its level may rise to approximately 4.0 µg/L without any postoperative neurological deficit [21,26]. Of note, the baseline value of S100 β in CSF was found to show considerable variation from person to person. Thus, it remains unknown whether it is the absolute or relative elevation of S100 β that should be considered when assessing its diagnostic potential for early prediction of severe CNS damage. 

Finally, a retrospective study by Gombert et al. compared the level of S100 β in CSF in patients undergoing OSR (14 patients) or coEVAR (19 patients) of TAAA. A decrease in amplitude of the motor evoked potentials (MEPs) by more than 50%, indicating SCI, was set in relation to S100 β measurements [27]. Additionally, the level of biomarkers was correlated with in-hospital mortality rates. SCI was diagnosed in three patients, two of which were treated with endovascular repair leading to paraplegia. A pathological decrease in MEPs could be observed in all symptomatic patients. No significant difference was found between patients treated with OSR versus coEVAR with regard to S100 β levels. S100 β showed a statistically significant increase over time in both, the OSR and coEVAR cohort (*p* = 0.0017 for combined data set). For coEVAR specifically, the highest elevation of S100 β was seen at 24 h after admission of a patient to an intensive care unit (ICU). In contrast to a previous study with patients undergoing OSR (where elevated S100 β was found to correlate with postoperative paraplegia and a decrease in the amplitude of transcranial MEPs [26]), Gombert and colleagues did not detect any association. At none of the investigated time-points did S100 β show a significant correlation with a decrease in intraoperative MEPs or with in-hospital mortality rates.

Gradual CSF S100 β elevation to a peak level at 24 h after procedure in both symptomatic and non-symptomatic groups of patients could be observed in all studies discussed above. Given a limited number of patients experiencing neurological deficit, no conclusions regarding diagnostic patency of S100 β could be made based on the evidence of present studies. At the same time, its assessment might potentially provide an insight into a specific time-point at which spinal cord is most severely exposed to ischemia following stent-graft deployment.

#### 2.1.2. GFAP (Glial Fibrillary Acidic Protein)

GFAP is the main type III intermediate filament protein in mature astrocytes that additionally represents an important component of the cytoskeleton in astrocytes during development [16,28]. Attributable to the discovery of novel intermediate filament functions and major revelations in astrocyte biology, GFAP was found to play a critical role in astrocyte regeneration, synaptic plasticity, axonal structural support, mechanical strength and reactive gliosis [28]. The latter is associated with a cascade of repair mechanisms directed to minimize damage after central nervous system injury, including the regulation of blood flow following CNS ischemia [29]. Considering this, GFAP constitutes a major biomarker of interest in both traumatic and ischemic spinal cord injury [30]. Its diagnostic potential for early stent-graft associated SCI was investigated in two studies.

In a pilot trial exploring the dynamics of selected biomarkers in CSF, including GFAP, amongst nine patients undergoing coEVAR, one patient developed spinal cord and supratentorial brain ischemia, as confirmed by MRI [15]. This patient had been treated by B-EVAR which included the revascularization of all visceral arteries. Neither the left subclavian artery nor the internal iliac artery were occluded prior to the procedure. In accordance with the MALAN criteria, aortic coverage was performed from zone one to nine, representing 98% of total aortic coverage [31]. In the majority of patients (5/9), an initially stable phase for the first postoperative hours was indicated by a trend for moderate GFAP elevation in comparison to the baseline value. GFAP concentrations showed a marked increase in symptomatic vs. non-symptomatic patients, which was most notably observed in samples taken 20–28 h after the induction of anaesthesia (95,708% for the symptomatic patient vs. 110–2011% increase for the non-symptomatic group). In addition to these findings, a biomarker potential of GFAP analysed in CSF for SCI has previously been supported by other studies performed on OSR, where GFAP was the only investigated marker significantly raised before the clinical onset of ischemic SCI [20].

In the above-mentioned study by Gombert et al., no statistically significant fluctuations over time were observed for GFAP [27]. The median increase of GFAP concentration at 24 h after admission to ICU reached 0.23 ng/mL vs. 0.15 ng/mL at baseline. Accordingly, the level of GFAP (at all time-points) in the three patients suffering from postoperative SCI was not associated with a decrease in MEP amplitude or in-hospital mortality as assessed in univariate logistic regression models.

According to S100 β, the findings of both studies implied the most remarkable elevation of CSF concentration of GFAP at 24 h after procedure. Following this, GFAP might potentially complement a panel of biomarkers reflecting a postoperative period at which patients are exposed to the most extensive ischemia-reperfusion damage. Nevertheless, considering the non-significant correlation of GFAP elevation and clinical outcome, no critical conclusions could be made with regard to its predictive nature of stent-graft-associated SCI. 

### 2.2. Markers of Axonal Injury (NFL (Neurofilament Light Chain) and Tau)

NFL is a neuronal cytoplasmic protein that is highly expressed in myelinated axons [32]. Similarly to NFL, the microtubule-stabilizing Tau protein is primarily localized in neurons and especially abundant in the axonal compartment [33]. Both proteins contribute to axonal transport and play an integral role in the neuronal transmission processes [34]. Under healthy conditions, NFL and Tau are not secreted into CSF or blood. Therefore, their detection was found to be indicative for various neuropathological conditions, including stroke or traumatic brain injury, in animal and human studies [35,36,37]. The role of NFL and Tau proteins as potential predictors of stent-graft associated SCI was assessed in two investigations. 

Merisson and colleagues sampled CSF for up to 48 h in ten patients undergoing coEVAR. Six procedures resulted in SCI which was clinically present within 3–6 h after intervention [38]. It should be noted that patients with SCI were treated for TAAA with branched devices and had a longer duration of intervention than those without SCI. The level of NFL and Tau did not vary between patients at baseline. In the six patients with clinical SCI, the postoperative phase was accompanied by an up to 37-fold increase of NFL concentrations at 24–48 h after the intervention. Although CSF levels of Tau were also elevated in this symptomatic group of patients by two to four-fold within 12–48 h, a comparable trend was observed in patients without any neurological deficits. Although promising, the level of NFL and Tau did not increase before appearance of the first clinical signs of SCI. 

The prospective observational study by Jónsson et al. demonstrated for a patient experiencing postoperative SCI and stroke that the CSF concentration of NFL increased to 467%, compared to 74–162% in eight asymptomatic patients [15]. For Tau, the corresponding increase was 1002% for the SCI patient compared to 100–454% in the non-symptomatic group. These peak values for both biomarkers were observed in samples taken 20–28 h after branched EVAR. 

Provided the high incidence of SCI in one of the studies discussed above, more concrete conclusions with regards to clinical application of NFL and Tau could be made. Although promising in terms of an inherent increase of both biomarkers in CSF of symptomatic patients, NFL and Tau are not suitable for early detection of neuronal injury since its elevation did not precede the appearance of first clinical signs of SCI. 

### 2.3. Glucose and Insulin Resistance 

The search for a predictive marker of ischemic SCI in patients with open or endovascular TAAA repair has essentially been based on evidence from studies with traumatic SCI or other acute neurological conditions [18,36,37,39]. Hyperglycaemia is considered to be a detrimental factor that impairs functional and clinical outcome in patients with acute ischemic stroke, subarachnoid haemorrhage or traumatic head injury [40,41,42,43,44,45,46]. Proposed pathological processes accounting for harmful effects of hyperglycaemia include oxidative injury, loss of selective membrane permeability and calcium homeostasis, free radical formation, and elevation of lactic acid [47,48,49,50,51]. In addition, animal studies suggest that acute hyperglycaemic conditions are associated with enhanced inflammatory responses, resulting from activation of the nuclear factor κB transcription factor in microglial cells [52].

With regard to stent-graft associated SCI, one study has examined the association between postoperative lower extremity weakness (LEW) and the glucose levels of CSF and blood [53]. 

#### 2.3.1. Glucose

In the study by Hiramoto et al., the assessment of glucose levels at three pre-defined time-points (baseline, immediately after coEVAR and 1 day after intervention) was accompanied by additional measurement of CSF lactate levels in 21 patients undergoing multibranched aortic repair of TAAA [53]. Two patients were excluded from the analysis because of pre-existing paraplegia resulting from prior OSR. In 37% of the patients (7/19) B-EVAR resulted in postoperative LEW, which remained persistent after discharge in two out of seven symptomatic study participants. It is important to mention that symptomatic and asymptomatic patient groups showed no differences regarding the prevalence of diabetes mellitus or peripheral artery disease, the operation time, blood loss or preoperative baseline glucose levels. Patients who experienced either transient or permanent LEW had a significantly higher level of glucose in peripheral blood (PB) and CSF at the first postoperative day in comparison to non-symptomatic patients (*p* = 0.01, *p* = 0.004, respectively). Postoperative lactate levels measured in CSF were also significantly increased in the LEW group, not only at the first postoperative day (*p* = 0.004) but also immediately after intervention (*p* = 0.02). Essentially, the rise in blood and CSF glucose levels preceded the onset of LEW, and it was identified as the only significant independent predictor associated with the development of LEW in multivariate logistic regression model (odds ratio, 2.30 [1.03–5.14] per 10 mg/dL increase in CSF glucose; *p* = 0.04). 

Following the conclusions of their study, suggesting that abnormal glucose metabolism might be a causative factor of ischemic SCI, the investigators changed their institutional SCI prevention protocol. Consequently, patients undergoing BEVAR were treated with an intravenous insulin infusion to maintain blood glucose levels <120 mg/dL for 48 h after surgery, which resulted in a significant reduction of the incidence of postoperative LEW [54]. 

The estimation of blood and CSF levels of glucose provides an interesting insight into pathophysiological processes involved in progressive SCI formation. At the same time, considering the non-specific nature of glucose level assessment, its alteration should rather be considered as a risk factor of neurological impairment. In contrast to biochemical markers arising as a consequence of neuronal tissue damage, it remains unknown whether hyperglycaemia direct exerts detrimental effects on neuronal tissue or is caused by severe injury of the spinal cord. Nevertheless, the time and costs effectiveness of glucose level assessment in clinical settings should not be underestimated. 

#### 2.3.2. Insulin Resistance

While increased glucose levels were demonstrated to predict postoperative LEW, the subsequent study by Hiramoto and colleagues addressed the pathological mechanisms leading to acute hyperglycaemia in patients undergoing BEVAR procedures [55]. It was hypothesised that postoperative hyperglycaemia might be associated with an acute insulin resistance within the CNS. Of note, previous neurological studies had shown that exosomes of neuronal origin may shape the insulin response and their circulating levels in PB may reflect in vivo neuropathological processes. For instance, insulin resistance in patients with Alzheimer’s disease was found to be accompanied by a diminished glucose uptake in the brain [56,57]. This was associated with increased levels of an alternatively phosphorylated type 1 insulin receptor substrate (IRS-1) originating from neuron-derived exosomes (NDEs). Thus, Hiramoto et al. aimed to investigate whether an acute reduction of glucose intake by the spinal cord might be due to changes in insulin receptor signalling. 

The explorative analysis of NDEs was performed in ten BEVAR patients with a range of postoperative LEW using PB drawn at three time-points (baseline, immediately after BEVAR and 1 day after intervention). The level of NDEs and their status of IRS-1 phosphorylation, known to reflect insulin resistance, were assessed [57,58,59]. 

It could be demonstrated that the clinical forms of LEW after BEVAR (two patients with transient LEW, and two patients with permanent LEW) significantly correlated with an acute state of insulin resistance indicated by neuronal exosomes. Specifically, the IRS-1 phosphorylation pattern of NDEs reflecting insulin resistance was increased five-fold immediately after intervention (*p* = < 0.001). Apart from showing an effect of extensive aortic coverage on the IRS-1 profile, BEVAR procedures were also associated with an acute postoperative decrease in the number of NDEs, in particular, within patients with postoperative LEW (*p* = 0.05). In this regard, it was postulated that the reduction of NDEs might be attributable to stressed ischemic neuron conditions, or potentially to increased distribution to other CSN cells as part of the exosomal system of intercellular communication. 

Although the assessment of NDEs reflects, to a certain extent, pathophysiological processes taking place directly in the CNS, its implementation as a potential risk estimator of SCI remains to be proven. It remains unknown whether acute hyperglycemia represents a causative or a consequent factor of neuronal tissue damage, which might constitute the aim of further research works. Whereas insulin resistance is widely recognized to play a key role in type 2 diabetes mellitus (T2DM), larger cohort of participants including those with T2DM should be enrolled in following studies [60]. 

### 2.4. Other Markers 

The recently published study by Jónsson et al. (mentioned above) further compared the dynamics of other biomarkers in CSF after coEVAR, which have not been assessed previously in the context of complex aortic interventions [15]. The panel selection was again based on the evidence of prior neurological studies. Soluble amyloid precursor proteins (APP) α and β have been at the center of Alzheimer’s disease research for many years [61]. Their elevation has been additionally detected in reversible neurological ischemic conditions [62,63]. The panel was complemented by the analyses of amyloid β 38, 40 and 42 (Aβ38, 40, 42) as representative markers of amyloidogenic APP processing. Reduced levels of Aβ38, 40, 42 may potentially serve as an indicator of reduced synaptic transmission, a process which was shown to occur secondary to abrupt hypoperfusion of the spinal cord [63]. The biomarker panel investigated by Jónsson et al. [15] also included chinitinase-3-like protein 1 (CHI3L1) which is a glycoprotein associated with tissue remodelling processes, inflammation and fibrosis, that was suggested as an important marker of astrocytic activation [29,64]. Furthermore, CSF samples were assessed for the presence of heart-type fatty acid binding protein (H-FABP), mostly known for its release from cardiac myocytes during ischemia, and which is also identified as a marker of neuronal injury [65,66]. As previously mentioned, this novel panel of biomarkers was complemented with the assessment of GFAP, NFL and Tau proteins [15].

The collection of CSF samples was performed up to 80 h after induction of anaesthesia in ten patients treated with coEVAR. After an initial stable phase for the first postoperative hours, all ten biomarkers showed an upward trend compared to baseline levels in the combined collective of the patients, including those with and without neurological deficits. Of note, a more than 50% increase could only be confirmed for GFAP, NFL and Tau proteins. One patient experienced fulminant SCI and stroke. Neurological symptoms were first observed on the second postoperative day after reduction of sedation, with paraplegia and left unilateral arm paralysis. In this patient, the most notable elevation of biomarkers occurred at 20–28 h. When assessing the differences between the symptomatic and asymptomatic group, it was only the previously described group of biomarkers (GFAP, NFL, Tau) that showed a significant increase in patients with SCI, while the panel of novel biomarkers did not exhibit diagnostic power in the present study.

Alongside with these findings, the authors proposed that a gradual elevation of biomarkers in asymptomatic patients might reflect the degree of adjustment processes of collateral network supply to the spinal cord secondary to stent-graft coverage of critical segmental arteries (Sas). However, considering the limited number of participants and the explorative design of the study, the future potential of biomarkers, particularly as a relevant parameter for subclinical forms of SCI, remains to be validated. 

## 3. Conclusions and Future Prospects

The reduction of spinal cord perfusion secondary to the occlusion of critical blood vessels upon TAAA repair causes a complex chain of pathological processes. Accordingly, extensive ischemic damage of nerve tracts in the spinal cord has been shown to be accompanied by an elevation of proteins and metabolites from nervous tissue, which are unique in their origin, therefore potentially representing sensitive biomarkers of SCI. Studies devoted to traumatic SCI established a specific set of biomarkers, which predicted the risk of neurological impairment and survival of the patients. The vast majority of evidence on the diagnostic potential of biomarkers in the context of TAAA repair originates from OSR studies. Since the pathophysiological processes of progressive SCI differ substantially between patients undergoing OSR and coEVAR, the current limitation for clinical translation is based on the fact that significantly less evidence exists with regard to biomarker application in patients undergoing EVAR. The present literature (as further summarised in Table 1 and Figure 1) is limited to studies performed in small collectives of patients, the majority of which did not show any postoperative neurological deficits. Following the above mentioned, more critical conclusions regarding diagnostic potential of biomarkers for early SCI could be made based on the evidence of study by Merisson et al. [38]. With 60% of the patients experiencing postoperative neurological deficit, neither the elevation of NFL or Tau preceded the appearance of first clinical signs of SCI. Therefore, it may be concluded that this panel of biomarkers is potentially not suitable for application upon clinical setting.

Nevertheless, based on the results of discussed literature some of the biomarkers show a uniform tendency for graduate elevation following stent-graft deployment in both symptomatic and non-symptomatic patients. For instance, it was the CSF level of S100 β, GFAP, NFL and Tau the peak value of which was most frequently observed at 24 h after procedure [15,24,25,27,38]. Although its potency as a biomarker of early SCI could not be confirmed in present studies, the dynamics of the aforementioned biomarkers contribute to our understanding of an exact postoperative period at which the spinal cord is potentially more severely exposed to ischemia-reperfusion injury. This offers a novel conception of clinical utilization of the biomarkers of which assessment might guide clinicians and surgeons to adjust neuroprotective measures at a certain time-point. 

Of note, coEVARs are potentially associated with other postoperative ischemic complications, e.g., mesenteric ischemia, which might also affect the biomarker levels. As discussed by most of the authors mentioned above, it remains to be established whether it is the relative or absolute level of biomarkers that is the more powerful predictor in patients with SCI. Although promising and logistically effective, the assessment of glucose levels should rather be considered as a non-specific estimator of postoperative neurological impairment. For instance, systemic inflammatory response syndrome has also been found to be accompanied by an acute hyperglycaemia, thus underestimating the specificity of its predictive features in terms of SCI [67]. Whereas the evaluation of NDEs in a study performed by Hiramoto et al. attempts to provide more neuronal tissue specific insight into alteration of glucose metabolism, it remains unknown whether its elevation represents a causative or a consequent factor of neuronal tissue damage [55]. Following this, the feasibility of glucose and insulin resistance markers application as either predictors or diagnostic parameters of postoperative SCI should be further evaluated in large cohort studies, including patients with acknowledged preoperative insulin resistance.

The evidence of discussed literature suggests that none of the biomarkers estimated in patients undergoing coEVAR is suitable enough to accurately assess early or subclinical forms of SCI. Provided the limited number of patients and the lack of patients experiencing postoperative neurological impairment, further prospective clinical studies are required to validate a biomarker for ischemic SCI. This would significantly add to the armamentarium of diagnostic modalities in patients undergoing coEVAR.

## Figures and Tables

**Figure 1 biomedicines-11-01317-f001:**
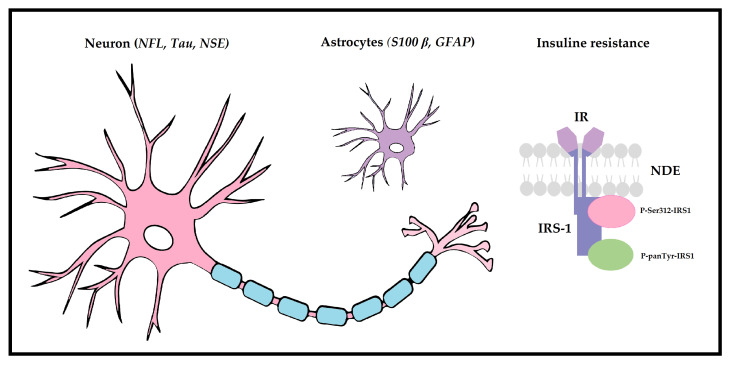
Overview of biomarkers investigated in patients with coEVAR. Neurofilament light chain (NFL), neuron specific enolase (NSE) and Tau protein represent markers of neuronal tissue damage. All of these factors contribute to axonal transport and play an integral role in neuronal transmission processes. S100 calcium-binding protein β (S100 β) and glial-fibrillary acidic protein (GFAP) assist in astrocyte regeneration processes, provide synaptic plasticity, axonal structural support, mechanical strength and are involved in reactive gliosis. Acute insulin resistance was found to be associated with an increased risk for postoperative lower extremity weakness (LEW): changes in the phosphorylation pattern of insulin receptor substrate 1 (IRS-1) as defined by the ratio of P-Ser312-IRS1 (leading to degradation of IRS-1 and inhibition of insulin signalling) to P-panTyr-IRS1 (promoting insulin stimulated responses) were evaluated on neuron derived exosomes (NDEs) and set in relation to LEW.

**Table 1 biomedicines-11-01317-t001:** Chronological summary of discussed studies assessing biomarkers of stent-graft associated SCI in TAAA and coAAA patients.

Study	Biomarkers	Sample Type	Sampling Schedule	Total Patients	Clinical Outcome	Biomarker Correlation
Brunnekreef et al. [25]	S100 β	CSF	After induction of anaesthesia, stent-graft deployment (SGD), 1, 2,3, 4, 5, 6, 10, and 48 h after SGD	8	No patient with SCI	No significant increase in S100 β; 3-fold increase in patient with the highest risk for SCI at 20 h after SGD placement versus baseline (<0.7 µg/L vs. 2.5 µg/L, respectively).
Merisson et al. [38]	T-tau, NFL	CSF, arterial blood	Before, after induction of anaesthesia, at 1, 3, 6, 12, 24 and 48 h after SGD	10	6/10 patients with persistent SCI (3/6–immediate SCI); At discharge: 5/6 with incomplete SCI; 1/6 with complete SCI	Elevation of CSF concentration of NFL up to 37-fold in patients with SCI; Elevation of T-Tau in CSF up to 4-fold in patients with SCI (overlap with patients without SCI).
Hiramoto J et al. [53]	Glucose, lactate	PB, CSF	Before BEVAR, immediately thereafter, 1st postoperative day	21	7/21 patients developed postoperative LEW (5/7: transient LEW, 2/7: permanent LEW)	Significant increase in glucose levels in PB and CSF in patients with LEW at the first postoperative day in comparison to non-symptomatic patients (*p* = 0.01, *p* = 0.004, respectively); Elevation of glucose levels preceded clinical symptoms.
Nandate et al. [24]	S100 β, antioxidant capacity	CSF	Before TEVAR, immediately after SGD, 24 and 48 h after SGD	10	No patient with SCI	Significant increase in S100 β 24 h after SGD vs. baseline (*p* = 0.007).
Gombert et al. [27]	NSE, GFAP, S100 β	CSF	Preoperatively, after CSFD placement, after admission to ICU, 12, 24, 48, and 72 h after intervention	19 coEVAR 14 OSR	3/33 patients with SCI (2/3 treated with EVAR–paraplegia, 1/3 treated with OSR–paraparesis)	No correlation with the end points “significant decrease in intraoperative MEPs measurement (>50%) and “in-hospital mortality”; Statistically significant difference for NSE and S100 β levels measured at different time-points (*p* = < 0.0001, *p* = 0.0017, respectively).
Hiramoto J et al. [55]	NDEs and IRS-1 phosphorylation profile	PB	Before BEVAR, immediately thereafter, 1st postoperative day	10	10 patients with various clinical grades of postoperative LEW: 2/10: permanent LEW 2/10: transient LEW	Significant increase in insulin resistance (as evidenced by IRS-1 phosphorylation pattern) immediately after BEVAR vs. baseline; Significant decrease in NDEs in patients with LEW.
Jónsson et al. [15]	NFL, Tau, GFAP, APPα, APPβ, Amyloidβ 38, 40, and 42, CHI3LI, H-FABP	CSF	During CSFD placement, after induction of anaesthesia, 0–5 min, 15 min, 30 min, 1, 3–6, 20–28, 44–52, and 72–80 h after SGD	9	1/9 patients with permanent SCI	Significant elevation of NFL, GFAP and Tau in patient with SCI at 20–28 h after SGP: Levels in SCI vs. no SCI patients: NFL: 467% vs. 74–162%; Tau: 1002% vs. 100–454%; GFAP: 95,708% vs. 110–2011%.

Abbreviations: TAAA, thoracoabdominal aortic aneurysm; coAAA, complex abdominal aortic aneurysm; S 100 β, S 100 calcium-binding protein β;, CSF, cerebrospinal fluid; SGD, stent-graft deployment; SCI, spinal cord ischemia; NFL, neurofilament light; PB, peripheral blood; BEVAR, branched endovascular aneurysm repair; LEW, lower extremity weakness; TEVAR, thoracic endovascular aneurysm repair; GFAP, glial fibrillary acidic protein; ICU, intensive care unit; coEVAR, complex endovascular aneurysm repair; OSR, open surgical repair; MEPs, motor-evoked potentials; NDE, neuron derived exosomes; IRS-1, type 1 insulin receptor substrate; APPα, amyloid precursor protein α; APPβ, amyloid precursor protein β; CHI3LI, chinitinase-3-like protein 1; H-FABP, Heart-type fatty acid binding protein.

## Data Availability

No new data were created or analysed in this review. Data sharing is not applicable to this article.

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
