# Peer review of "Biomarkers of Spinal Cord Injury in Patients Undergoing Complex Endovascular Aortic Repair Procedures—A Narrative Review of Current Literature"

_biomedicines, 2023, doi:10.3390/biomedicines11051317_

Round 1

Reviewer 1 Report

Specify in the title that is a review

Introduction: specify type and method of the review before describing all markers

Current perspective and future prospects has to be improved, this section is very poor. Insert more comment and considerations

Author Response

Response to Reviewer 1 Comments

We thank the Reviewer for all the valuable comments and suggestions.

Point 1: Specify in the title that is a review

Response 1: We have updated the title of the manuscript accordingly. “Biomarkers of spinal cord injury in patients undergoing complex endovascular aortic repair procedures – a narrative review of current literature” Please, refer to the title page, Line 2-4.

Point 2: Introduction: specify type and method of the review before describing all markers

Response 2: We thank the Reviewer for this comment. We have specified in introduction section, that this was a narrative review, and have included information regarding literature search database and keywords used. “This narrative review discusses published literature on the diagnostic potential of biomarkers of SCI in the context of coEVAR. The cohort of interest included patients treated for complex abdominal aortic aneurysms (coAAA) or TAAA with thoracic, fenestrated and/or branched stent-grafts (T/F/B-EVAR). PubMed database was used to perform literature search. The search string included the combination of the following keywords: “spinal cord injury”, “biomarkers”, “biochemical markers”, “thoracoabdominal aortic aneurysm”, “complex abdominal aortic aneurysm”, “endovascular aortic repair”, “fenestrated endovascular aortic repair”, “branched endovascular aortic repair”, “S100 β”, “GFAP”, “NFL”, “Tau”, “lactate”. ” Please, refer to Page 2, Line 72-80.

Point 3: Current perspective and future prospects has to be improved, this section is very poor. Insert more comment and considerations

Response 3: We certainly agree with a Reviewer that our conclusion section should have included more concrete statements with regards to potential application of discussed biomarkers in clinical setting.

Therefore, we have supplemented our conclusion section with the following comments.

“The present literature (as further summarised in Table 1 and Figure 1) is limited to studies performed in small collectives of patients, the majority of which did not show any postoperative neurological deficits. Following the above mentioned, more critical conclusions regarding diagnostic potential of biomarkers for early SCI could be made based on the evidence of study by Merisson et al. [39]. With 60% of the patients experiencing postoperative neurological deficit, neither the elevation of NFL or Tau preceded the appearance of first clinical signs of SCI. Therefore, it may be concluded that this panel of biomarkers is potentially not suitable for application upon clinical setting. Nevertheless, based on the results of discussed literature some of the biomarkers show a uniform tendency for graduate elevation following stent-graft deployment in both symptomatic and non-symptomatic patients. For instance, it was the CSF level of S100 β, GFAP, NFL and Tau the peak value of which was most frequently observed at 24 hours after procedure [15,25,26,28,39]. Although its patency as a biomarker of early SCI could not be confirmed in present studies, the dynamics of aforementioned biomarkers contribute to our understanding of an exact postoperative period at which spinal cord is potentially more severely exposed to ischemia-reperfusion injury. This offers a novel conception of clinical utilization of biomarkers which assessment might guide clinicians and surgeons to adjust neuroprotective measures at a certain time-point.” Please, refer to Page 8, Line 362-380. 

“Although promising and logistically effective, the assessment of glucose levels should rather be considered as a non-specific estimator of postoperative neurological impairment. For instance, systemic inflammatory response syndrome has also been found to be accompanied by an acute hyperglycaemia, thus underestimating the specificity of its predictive features in terms of SCI[68]. Whereas the evaluation of NDEs in a study performed by Hiramoto et al. attempts to provide more neuronal tissue specific insight into alteration of glucose metabolism, it remains to be unknown whether its elevation represents a causative or a consequent factor of neuronal tissue damage[56]. Following this, the feasibility of glucose and insulin resistance markers application as either predictors or diagnostic parameters of postoperative SCI should be further evaluated in large cohort studies, including patients with acknowledged preoperative insulin resistance.” Please, refer to Page 8, Line 385-396.

“The evidence of discussed literature suggests that none of the biomarkers estimated in patients undergoing coEVAR is suitable enough to accurately assess early or subclinical forms of SCI. Provided the limited number of patients and the lack of patients experiencing postoperative neurological impairment”. Please, refer to Page 8, Line 397-401.

Reviewer 2 Report

The manuscript under revision provides a brief overview of the potential role of endogenous compounds specific to neuronal tissue damage as predictive/ risk stratification biomarkers for spinal cord injury (SCI) in patients undergoing complex endovascular aortic repair procedures (co-EVAR).

The topic is interesting and actual, SCI being a rare but devastating complication of EVAR. At the same time, clinical research in SCI is difficult to be conducted, due to its relatively low incidence and heterogeneity.

Comments

Major

1.       The paucity of clinical studies included in this review is understandable, due to the above mentioned limitations; though, the content is rather descriptive, therefore it is highly advisable to include pertinent  and robust commentaries of the presented data.

2.       At the end of each section presenting a specific marker, please add a critical conclusion regarding its relevance/potential as a predictive biomarker for SCI or as a risk stratification element.

3.       Section 2.3. consists entirely of a narration of Jónsson et al. study (ref. 15) – would be recommended to refer to additional sources that validated the tested markers in similar circumstances, as a starting point for commenting their value as potential markers for SCI in coEVAR

4.       In the ‘Current perspective and future prospects’ part, one expects to find a concluding accolade that points out on which ones of the discussed markers are most promising as a predictor/ risk stratification item, why it was/ or was not relevant enough, and offers potential clues for future trials etc.

Minor

1.       Since the review is focused on SCI post-coEVAR, please clarify on which basis you have included the various types of EVAR cited (TEVAR, TAAA, B-EVAR etc) in the coEVARs category.

2.      Section 2.3.: High glucose and insulin resistance are both rather risk factors than biomarkers

3.       Table 1 – TAAA abbreviation is not explained

4.       Subchapter numbering, a suggestion: '2. Biochemical markers of .. '– the markers discussed are biochemicals, therefore may be more appropriate to either name it ‘Markers of…’, or to count the subsequent sections as 2.‘Biomarkers of astrocyte activation’ 3. Bioamarkers of axonal injury etc.

Minor editing of English language is required

Author Response

Response to Reviewer 2 Comments

We thank the Reviewer for all the valuable suggestions and comments.

Major

Point 1: The paucity of clinical studies included in this review is understandable, due to the above mentioned limitations; though, the content is rather descriptive, therefore it is highly advisable to include pertinent and robust commentaries of the presented data.

Response 1: We certainly agree with a Reviewer. Therefore, at the end of each section devoted to a specific biomarker we have added a critical conclusion regarding its diagnostic potential for early SCI diagnosis or further possibilities of its clinical application.

  • S100 β (S100 calcium-binding protein β)

“Graduate CSF S100 β elevation to a peak level at 24 hours after procedure in both symptomatic and non-symptomatic groups of patients could be observed in all studies discussed above. Given a limited number of patients experiencing neurological deficit, no conclusions regarding diagnostic patency of S100 β could be made based on the evidence of present studies. At the same time, its assessment might potentially provide an insight into a specific time-point at which spinal cord is most severely exposed to ischemia following stent-graft deployment.” Please, refer to Page 3, Line 134-140.

  • GFAP (glial fibrillary acidic protein)

“According to S100 β, the findings of both studies implied the most remarkable elevation of CSF concentration of GFAP at 24 hours after procedure. Following this, GFAP might potentially complement a panel of biomarkers reflecting a postoperative period at which patients are exposed to the most extensive ischemia-reperfusion damage. Nevertheless, considering the non-significant correlation of GFAP elevation and clinical outcome, no critical conclusions could be made with regards to its predictive nature of stent-graft associated SCI.” Please, refer to Page 4, Line 176-182.

  • NFL (neurofilament light chain) and Tau

“Provided the high incidence of SCI in one of the studies discussed above, more concrete conclusions with regards to clinical application of NFL and Tau could be made. Although promising in terms of an inherent increase of both biomarkers in CSF of symptomatic patients, NFL and Tau are not suitable for early detection of neuronal injury since its elevation did not precede the appearance of first clinical signs of SCI.” Please, refer to Page 5, Line 214-218.

  • Glucose

The estimation of blood and CSF levels of glucose provides an interesting insight into pathophysiological processes involved in progressive SCI formation. At the same time, considering the non-specific nature of glucose level assessment, its alteration should rather be considered as a risk factor of neurological impairment. On contrast to biochemical markers arising as a consequence of neuronal tissue damage, it remains unknown whether hyperglycaemia direct exerts detrimental effects on neuronal tissue or is caused by severe injury of the spinal cord. Nevertheless, time and costs effectiveness of glucose level assessment in clinical setting should not be underestimated. Please, refer to Page 6 , Line 263-270.

  • Insulin resistance

“Although, the assessment of NDEs reflects, to a certain extent, pathophysiological processes taking place directly in the CNS, its implementation as a potential risk estimator of SCI remains to be proven. It remains unknown whether acute hyperglycemia represents a causative or a consequent factor of neuronal tissue damage, which might constitute the aim of further research works. Whereas insulin resistance is widely recognized to play a key role in type 2 diabetes mellitus (T2DM), larger cohort of participants including those with T2DM should be enrolled in following studies [61].” Please, refer to Page 6, Line 301-303 and Page 7, Line 304-308.

Point 2: At the end of each section presenting a specific marker, please add a critical conclusion regarding its relevance/potential as a predictive biomarker for SCI or as a risk stratification element.

Response 2: We thank the Reviewer for giving us this valuable suggestion. Each section of the manuscript presenting a specific biomarker was complemented with critical conclusion regarding its relevance or future potential for early SCI diagnosis/risk prediction.

  • S100 β (S100 calcium-binding protein β)

“Graduate CSF S100 β elevation to a peak level at 24 hours after procedure in both symptomatic and non-symptomatic groups of patients could be observed in all studies discussed above. Given a limited number of patients experiencing neurological deficit, no conclusions regarding diagnostic patency of S100 β could be made based on the evidence of present studies. At the same time, its assessment might potentially provide an insight into a specific time-point at which spinal cord is most severely exposed to ischemia following stent-graft deployment.” Please, refer to Page 3, Line 134-140.

  • GFAP (glial fibrillary acidic protein)

“According to S100 β, the findings of both studies implied the most remarkable elevation of CSF concentration of GFAP at 24 hours after procedure. Following this, GFAP might potentially complement a panel of biomarkers reflecting a postoperative period at which patients are exposed to the most extensive ischemia-reperfusion damage. Nevertheless, considering the non-significant correlation of GFAP elevation and clinical outcome, no critical conclusions could be made with regards to its predictive nature of stent-graft associated SCI.” Please, refer to Page 4, Line 176-182.

  • NFL (neurofilament light chain) and Tau

“Provided the high incidence of SCI in one of the studies discussed above, more concrete conclusions with regards to clinical application of NFL and Tau could be made. Although promising in terms of an inherent increase of both biomarkers in CSF of symptomatic patients, NFL and Tau are not suitable for early detection of neuronal injury since its elevation did not precede the appearance of first clinical signs of SCI.” Please, refer to Page 5, Line 214-218.

  • Glucose

The estimation of blood and CSF levels of glucose provides an interesting insight into pathophysiological processes involved in progressive SCI formation. At the same time, considering the non-specific nature of glucose level assessment, its alteration should rather be considered as a risk factor of neurological impairment. On contrast to biochemical markers arising as a consequence of neuronal tissue damage, it remains unknown whether hyperglycaemia direct exerts detrimental effects on neuronal tissue or is caused by severe injury of the spinal cord. Nevertheless, time and costs effectiveness of glucose level assessment in clinical setting should not be underestimated. Please, refer to Page 6 , Line 263-270.

  • Insulin resistance

“Although, the assessment of NDEs reflects, to a certain extent, pathophysiological processes taking place directly in the CNS, its implementation as a potential risk estimator of SCI remains to be proven. It remains unknown whether acute hyperglycemia represents a causative or a consequent factor of neuronal tissue damage, which might constitute the aim of further research works. Whereas insulin resistance is widely recognized to play a key role in type 2 diabetes mellitus (T2DM), larger cohort of participants including those with T2DM should be enrolled in following studies [61].” Please, refer to Page 6, Line 301-303 and Page 7, Line 304-308.

Point 3: Section 2.3. consists entirely of a narration of Jónsson et al. study (ref. 15) – would be recommended to refer to additional sources that validated the tested markers in similar circumstances, as a starting point for commenting their value as potential markers for SCI in coEVAR.

Response 3: At the beginning of section 2.3., we briefly described the panel of biomarkers assessed with respect to stent-graft associated SCI. In addition, we have cited and outlined studies which evaluated the biomarker of choice in other neuropathological conditions. The panel selection was again based on the evidence of prior neurological studies. Soluble amyloid precursor proteins (APP) a and b have been at the center of Alzheimer’s disease research for many years [62]. Their elevation has been additionally detected in reversible neurological ischemic conditions [63,64]. The panel was complemented by the analyses of amyloid b 38, 40 and 42 (Ab38, 40, 42) as representative markers of amyloidogenic APP processing. Reduced levels of Ab38, 40, 42 may potentially serve as an indicator of reduced synaptic transmission, a process which was shown to occur secondary to abrupt hypoperfusion of the spinal cord [64]. The biomarker panel investigated by by Jónsson et al.[15] also included chinitinase-3-like protein 1 (CHI3L1) which is a glycoprotein associated with tissue remodelling processes, inflammation and fibrosis, that was suggested as an important marker of astrocytic activation [30,65]. Furthermore, CSF samples were assessed for the presence of heart-type fatty acid binding protein (H-FABP), mostly known for its release from cardiac myocytes during ischemia, and which is also identified as a marker of neuronal injury [66,67].Please, refer to Page 7, Line 313-327.

As far as we are concerned, the panel of novel biomarkers assessed in a study by Jónsson et al. was not evaluated in other studies related to SCI as a complication of either EVAR or OSR of TAAA or complex AAA. Therefore, the results of discussed study could not be critically assessed in comparison to studies that validated the tested biomarkers in similar circumstances.

Point 4: In the ‘Current perspective and future prospects’ part, one expects to find a concluding accolade that points out on which ones of the discussed markers are most promising as a predictor/ risk stratification item, why it was/ or was not relevant enough, and offers potential clues for future trials etc.

Response 4: We certainly agree with a Reviewer that our conclusion section should have included more concrete statements with regards to potential application of discussed biomarkers in clinical setting. Therefore, we have supplemented our conclusion section with the following comments.

“The present literature (as further summarised in Table 1 and Figure 1) is limited to studies performed in small collectives of patients, the majority of which did not show any postoperative neurological deficits. Following the above mentioned, more critical conclusions regarding diagnostic potential of biomarkers for early SCI could be made based on the evidence of study by Merisson et al. [39]. With 60% of the patients experiencing postoperative neurological deficit, neither the elevation of NFL or Tau preceded the appearance of first clinical signs of SCI. Therefore, it may be concluded that this panel of biomarkers is potentially not suitable for application upon clinical setting. Nevertheless, based on the results of discussed literature some of the biomarkers show a uniform tendency for graduate elevation following stent-graft deployment in both symptomatic and non-symptomatic patients. For instance, it was the CSF level of S100 β, GFAP, NFL and Tau the peak value of which was most frequently observed at 24 hours after procedure [15,25,26,28,39]. Although its patency as a biomarker of early SCI could not be confirmed in present studies, the dynamics of aforementioned biomarkers contribute to our understanding of an exact postoperative period at which spinal cord is potentially more severely exposed to ischemia-reperfusion injury. This offers a novel conception of clinical utilization of biomarkers which assessment might guide clinicians and surgeons to adjust neuroprotective measures at a certain time-point.” Please, refer to Page 8, Line 362-380. 

“Although promising and logistically effective, the assessment of glucose levels should rather be considered as a non-specific estimator of postoperative neurological impairment. For instance, systemic inflammatory response syndrome has also been found to be accompanied by an acute hyperglycaemia, thus underestimating the specificity of its predictive features in terms of SCI[68]. Whereas the evaluation of NDEs in a study performed by Hiramoto et al. attempts to provide more neuronal tissue specific insight into alteration of glucose metabolism, it remains to be unknown whether its elevation represents a causative or a consequent factor of neuronal tissue damage[56]. Following this, the feasibility of glucose and insulin resistance markers application as either predictors or diagnostic parameters of postoperative SCI should be further evaluated in large cohort studies, including patients with acknowledged preoperative insulin resistance.” Please, refer to Page 8, Line 385-396.

“The evidence of discussed literature suggests that none of the biomarkers estimated in patients undergoing coEVAR is suitable enough to accurately assess early or subclinical forms of SCI. Provided the limited number of patients and the lack of patients experiencing postoperative neurological impairment”. Please, refer to Page 8, Line 397-401.

Minor

Point 1: Since the review is focused on SCI post-coEVAR, please clarify on which basis you have included the various types of EVAR cited (TEVAR, TAAA, B-EVAR etc) in the coEVARs category.

Response 1: We thank the Reviewer for this valuable suggestion. We have specified in the introduction section on which basis the literature search have been performed, including the details for the cohort of interest. “The cohort of interest included patients treated for complex abdominal aortic aneurysms (coAAA) or TAAA with thoracic, fenestrated and/or branched stent-grafts (T/F/B-EVAR). PubMed database was used to perform literature search. The search string included the combination of the following keywords: “spinal cord injury”, “biomarkers”, “biochemical markers”, “thoracoabdominal aortic aneurysm”, “complex abdominal aortic aneurysm”, “endovascular aortic repair”, “fenestrated endovascular aortic repair”, “branched endovascular aortic repair”, “S100 β”, “GFAP”, “NFL”, “Tau”, “lactate”.”  Please, refer to Page 2, Line 73-80.

Point 2: Section 2.3.: High glucose and insulin resistance are both rather risk factors than biomarkers

Response 2: Considering the non-specific nature of hyperglycaemia and insulin resistance, we certainly agree with a Reviewer that these parameters should rather be considered as risk factors of SCI. We have supplemented section 2.3 with the abovementioned conclusion.

“At the same time, considering the non-specific nature of glucose level assessment, it´s alteration should rather be considered as a risk factor of neurological impairment. On contrast to biochemical markers arising as a consequence of neuronal tissue damage, it remains unknown whether hyperglycaemia direct exerts detrimental effects on neuronal tissue or is caused by severe injury of the spinal cord.” Please, refer to Page 6, Line 264-269.

Although, the assessment of NDEs reflects, to a certain extent, pathophysiological processes taking place directly in the CNS, its implementation as a potential risk estimator of SCI remains to be proven. Please, refer to Page 6, Line 301-303.

Point 3: Table 1 – TAAA abbreviation is not explained

Response 3: We have included the explanation of TAAA abbreviation in “Abbreviations” section of Table 1. “Abbreviations: TAAA, thoracoabdominal aortic aneurysm;” Please, refer to Page 10, Line 406.

In addition, we have specified in the title of Table 1, that literature search was also performed for studies with patients treated for complex abdominal aortic aneurysm (coAAA). “Table 1. Chronological summary of discussed studies assessing biomarkers of stent-graft associated SCI in TAAA and coAAA patients.” Please, refer to Page 9, Line 404-405.

Point 4: Subchapter numbering, a suggestion: '2. Biochemical markers of .. '– the markers discussed are biochemicals, therefore may be more appropriate to either name it ‘Markers of…’, or to count the subsequent sections as 2.‘Biomarkers of astrocyte activation’ 3. Bioamarkers of axonal injury etc.

Response 4: We have updated the heading of section 2 with “Markers of stent-graft associated SCI”. Please, refer to Page 2, Line 82.

Round 2

Reviewer 1 Report

The corrections made are sufficient

no comment

Author Response

Dear Reviewer, 

Thank you for the information provided. 

Reviewer 2 Report

The comments and suggestions have been in their large majority correctly considered

Author Response

(The authors gave the same response as above.)
